# Evaluating the Agreement and Associations with Physical Function Between Equation- and Linear Position Transducer-Estimated Sit-to-Stand Muscle Power in Aging Adults

**DOI:** 10.3390/healthcare13080905

**Published:** 2025-04-15

**Authors:** Garrett Steinbrink, Taylor Danielson, Julian Martinez, Joseph Patnode, Ann Swartz, Scott Strath

**Affiliations:** Zilber College of Public Health, University of Wisconsin-Milwaukee, Milwaukee, WI 53211, USA; danie326@uwm.edu (T.D.); marti994@uwm.edu (J.M.); patnode@uwm.edu (J.P.); aswartz@uwm.edu (A.S.)

**Keywords:** strength, exercise, resistance training, older adults, physical activity, sarcopenia, frailty, primary care, gerontology

## Abstract

**Background/Objectives**: Muscle power, estimated from the sit-to-stand (STS) test, is an important indicator of physical function (PF) in aging adults. Therefore, its assessment may be implemented into future clinical practice. The agreement between different STS power assessments is unknown, and the associations between methods and PF outcomes have not been compared. **Methods**: A total of 49 aging adults (mean age = 60.9 ± 10.9; 67% female) participated in this cross-sectional study. STS power from a validated equation (EQ) and a linear position transducer (LPT) were estimated. Handgrip strength (HGS), timed up-and-go (TUG), usual gait speed (UGS), fast gait speed (FGS), the 400-m walk test (400MWT), and self-reported total, basic lower-body, and advanced lower-body PF were assessed. The agreement of STS power methods was assessed with an intraclass correlation coefficient (ICC) and a Bland–Altman plot. Multiple linear regression evaluated the associations between STS power and PF outcomes. **Results**: EQ and LPT STS power demonstrated only moderate agreement (ICC = 0.69). EQ STS power was independently associated with TUG (β = −0.45), UGS (β = 0.37), FGS (β = 0.48), 400MWT (β = −0.55), self-reported total (β = 0.30), basic lower-body (β = 0.30), and advanced lower-body PF (β = 0.30), but not HGS (β = 0.14). LPT STS power was independently associated with HGS (β = 0.44), FGS (β = 0.40), 400MWT (β = −0.51), self-reported total (β = 0.31), basic lower-body (β = 0.29), and advanced lower-body PF (β = 0.32), but neither TUG (β = −0.26) nor UGS (β = 0.28). **Conclusions**: EQ and LPT STS power demonstrate limited agreement, and EQ STS power may be a superior indicator of PF in aging adults. Future research should examine the feasibility of implementing STS power tests in clinical settings to screen and refer patients with low muscle power to effective therapeutic interventions.

## 1. Introduction

Muscle power is an important indicator of physical function (PF), mobility, and mortality in aging adults [1,2]. Muscle power is a stronger predictor of gait performance, functional mobility, and falls in aging adults compared to muscle strength [3,4,5,6,7]. Additionally, muscle power significantly predicts all-cause mortality, irrespective of sedentary behavior, physical activity, and adiposity [8], suggesting it should be treated as a critical marker of healthy aging and longevity [9]. As the number of adults 65+ years of age is expected to double in the coming decades, the frequent and accurate assessment of muscle power may play an important role in routine clinical practice.

One of the limitations in assessing muscle power is the accessibility of valid, easy-to-implement, and safe measurement methods [1]. The most commonly used instrument to quantify muscle power in aging adults is the pneumatic leg press, which is largely infeasible for the clinical setting [10]. Due to these methodological limitations, various research groups have developed equations (EQs) to estimate muscle power from the sit-to-stand (STS) test [1,11]. Of the available EQs, the EQ developed by Alcazar and colleagues is most strongly related to slow gait speed (i.e., <0.8 m/s) and frailty [11]. Moreover, the Alcazar EQ predicts numerous physical health-related variables, such as usual gait speed (UGS) and fast gait speed (FGS), timed up-and-go (TUG) performance, health-related quality of life, hospitalizations, and mortality in aging populations [8,11,12]. Importantly, EQ STS power is sensitive to intervention [13,14], and its improvement is significantly associated with augmented PF [14]. Therefore, EQ STS power could be used as a clinically useful, valid, and sensitive method to monitor aging adults’ muscle power.

While some research groups have developed EQs to estimate STS power, others have developed instrumented protocols using force plates [15,16], wearable monitors/accelerometers [17,18], and linear position transducers (LPTs) [5,19,20]. LPTs are relatively affordable, portable, and have the benefit of providing instantaneous, objective feedback to both patient and provider, with minimal data processing. Of clinical importance, LPT STS power is significantly associated with the Short Physical Performance Battery, gait speed, TUG performance, and balance in aging adults [20,21,22].

Multiple cut-off values for EQ STS power have recently been proposed and are associated with important aging outcomes such as sarcopenia, physical independence, mobility and frailty, ability to perform basic (BADL) and instrumental activities of daily living (IADL), hospitalizations, and poor quality of life [12,23,24,25]. For example, in a large sample of aging Spanish adults, EQ STS power cut-off values of 1.7 and 2.2 W/kg for females and males, respectively, have been proposed, and are associated with a greater likelihood of having poor PF (i.e., slow UGS, frailty, and limitations in ADLs and IADLs) [26]. Based on these proposed cut-off values, an operational clinical algorithm was developed to screen and manage the care of patients with low EQ STS power [26]. While not identifying cut-off values for low STS power, nor linking these data to PF outcomes, recently reported normative data in U.S. adults, using the TENDO LPT, report seemingly higher estimates of STS power [27] compared to the EQ STS power estimated in similar populations [23,25,26]. This suggests that STS power ascertained from different methods may yield considerably different estimates. Therefore, caution should be used when qualitatively describing patients’ STS power to make clinical decisions.

Discrepancies of STS power estimates between methods could be due to differences in test procedures. Specifically, EQ STS power is estimated from five consecutive STS transitions [1], whereas LPT STS power is assessed from one maximal effort stand from a seated position [20,27]. For the future screening and management of low muscle power in the clinical setting, it is important to establish the agreement between different methods. Doing so will determine whether proposed cut-off values can be applied across methods, or if method-specific cut-off values are required. Moreover, evaluating whether STS power methods differ in their relationships with PF outcomes in aging adults is important from a method selection perspective. Importantly, some evidence suggests that EQ STS power may predict TUG performance better than more objective, technology-based assessments [15,28,29]. However, to the authors’ knowledge, no study has directly compared the strengths of the relationships between different STS power methods and multiple objective and self-reported PF outcomes. Therefore, limited conclusions can be made regarding the relative ability of these methods to predict PF in aging adults.

The aims of this study, therefore, were to assess the absolute agreement between EQ and LPT STS power and to compare the relationships between EQ and LPT STS power assessments with objective and self-reported PF in aging adults. Based on previous estimates [26,27], we hypothesized that EQ and LPT STS power will showcase limited absolute agreement, with LPT STS power demonstrating consistently higher estimates than EQ STS power. Furthermore, based on limited, indirect evidence [15,29], we hypothesized that EQ STS power may demonstrate stronger relationships with PF outcomes compared to LPT STS power. The results of this study may have important implications for the future screening and management of low muscle power in the clinical setting.

## 2. Materials and Methods

### 2.1. Participants

Participants were recruited for this cross-sectional study via print and digital flyers, posted throughout various neighborhoods and businesses near a large, midwestern, urban university in the United States, and online via social media posts. Participants were eligible to participate in the study if they (1) were 40+ years of age, (2) could stand from a chair once and walk 10 m without the assistance of another person, and (3) could speak and read English. Participants were excluded from the study if they (1) used a wheelchair for mobility purposes, (2) were unable to safely follow the study procedures, or (3) had a resting heart rate or blood pressure over 100 bpm or 160/100 mmHg, respectively. Participants did not receive any financial compensation for their participation in this study, and all study procedures were reviewed and approved by the University of Wisconsin-Milwaukee’s institutional review board (protocol #23.133 on 6 January 2023). Participant recruitment and data collection for this study occurred between March 2023 and April 2024.

### 2.2. Sociodemographic Variables, Anthropometrics, and Body Composition

Sociodemographic and health history information were collected with a study-specific questionnaire. Participant body height and mass were measured with a calibrated stadiometer and physician’s scale, respectively (Detecto, Webb City, MO, USA). Body mass index (BMI) was calculated as body mass to height, squared (kg/m^2^).

### 2.3. STS Power Assessments

Randomization of the STS test order was performed by the lead author (G.S.) in February 2023, before beginning data collection. Both STS tests were completed on a 0.45 m-tall armless chair. For both assessments, participants began seated in the middle of the chair, with their feet flat on the floor and arms crossed over their chest, to limit the use of their upper extremities. To maintain ecological validity, no instruction was given with respect to starting ankle or knee joint angles [27].

To calculate EQ STS power, the total time for participants to perform 5 consecutive STS transitions, as quickly as possible, was measured to the nearest hundredths of a second with a manual stopwatch. The timing of the test began after the researcher said, “Ready, set, go!”, and stopped when the participant was seated on the fifth repetition [15]. The following validated EQ was used to estimate relative mean STS power [1]:(1)$$Absolute\;STS\;mean\;power\;\left(W\right)=\frac{Body\;mass\times 0.9\times g\times \left[\mathrm{Body}\;height\times 0.5-\mathrm{Chair}\;height\right]}{\mathrm{Five}\;\mathrm{STS}\;time\times 0.1}$$(2)$$Relative\;STS\;mean\;power\;\left(\frac{W}{kg}\right)=\frac{Absolute\;STS\;mean\;power}{Body\;mass}$$
where body mass is entered in kilograms, *g* is the gravitational constant 9.81, body height and chair height are entered in meters, and the time to complete five STS transitions are entered as seconds.

The LPT STS test was conducted as described by Sherwood and colleagues [20]. Briefly, the GymAware LPT, a valid and reliable LPT to assess STS power in aging adults [20], was positioned on a metal plate situated 8 cm lateral and 36 cm anterior to the back leg of the armless chair, and the unit’s tether was attached to the participant’s hip via a gait belt. Due to the feet and shank regions not being displaced during the STS transition, 90% of the participant’s body mass was entered into the GymAware iPad application (GymAware 3.2.3; Kinetic Performance Technology, Braddon, ACT, Australia) prior to the test. After the researcher explained the test protocol, participants were told to stand, as quickly as possible, from the chair a total of 5 separate times, with 1 min of rest given between each stand [20]. Mean and peak instantaneous velocity, force, and power parameters were estimated from each trial and later exported from the GymAware Cloud Pro software (https://gymaware.com/shop/#gymaware-cloud (accessed on 6 January 2023)). The maximal relative mean STS power elicited during the test was used for analysis.

### 2.4. Objective Physical Function Outcomes

Handgrip strength (HGS) is a proxy of overall muscle strength and is used to diagnose sarcopenia, or the age-related loss of skeletal muscle mass, strength, and function [30]. Participants sat on an adjustable chair, with their forearm resting on an adjustable table and their elbow joint flexed at approximately 90 degrees. The chair and table’s heights were adjusted to ensure the participants’ shoulders were level. After familiarization, participants were instructed to squeeze a calibrated hand dynamometer (Lafayette Professional Hand Dynamometer FE-12-5203, Lafayette Instrument, Lafayette, IN, USA) as hard as possible for five seconds. After each trial, participants were given 30 s of rest. Participants completed three trials in each hand, with the first set of trials being completed with the participants’ dominant hand. The maximal HGS attained over the six total trials was normalized to the participants’ body mass and used for analysis.

The TUG test is a functional mobility test and is an indicator of balance, gait speed, and ADL performance in the aging population [31]. Briefly, participants started the test seated in a standard armchair 0.46 m in seat height and 0.65 m in arm rest height. Participants were instructed to stand from the chair, walk around a cone situated 3 m away, return to the chair, and sit back down at their usual pace. The time taken to complete the test was measured on a stopwatch, measured to the nearest hundredths of a second.

The 10 m walk test is a valid and reliable assessment of gait speed in the adult population [32]. Gait speed is a potent predictor of mortality, mobility disability, and falls, and is often cited as the “functional vital sign” [33,34,35,36]. Given that UGS and FGS differ in their relationships to PF in community-dwelling aging adults [37], we assessed gait speed at both velocities. For the assessment of UGS, participants were instructed to walk 10 m down an unobstructed corridor at their usual, comfortable pace, while for the assessment of FGS, participants were instructed to walk the same distance at their maximal walking pace, without running. To account for acceleration and deceleration, the timed portion of the test included only the intermediate 6 m of the 10 m track. A researcher timed the participants with a manual stopwatch, to the nearest hundredths of a second, and UGS and FGS were estimated for each trial. Participants completed 3 trials of both UGS and FGS, and the average of the 3 trials was used for analysis.

The fast-paced 400 m walk test (400MWT) is a valid predictor of aerobic fitness in aging adults [38]. Each 1 min longer 400MWT time is associated with a ~30% increase in all-cause mortality [39], and the inability to complete the test is associated with a near two-fold greater risk of future mobility limitations and disability [40]. To complete the test, two cones were placed 20 m apart in an unobstructed corridor. Participants were instructed to complete 10 laps around the 40 m track, walking at their maximal pace, without running. Participants were allowed to stop twice throughout the test, if needed, but were told that the time would continue during rest periods. Time taken to complete the test was measured to the nearest second with a manual stopwatch.

### 2.5. Self-Report Physical Function

The Late-Life Function and Disability Instrument (LLFDI) is a valid self-report measure of PF in aging adults, with moderately strong relationships demonstrated between LLFDI subscales and objective assessments of function (i.e., SPPB and 400MWT performance) [41]. For this study, the functional domain of the LLFDI contains the primary outcomes of interest. Specifically, the scaled scores (0–100) of total, basic lower-body, and advanced lower-body function were the outcomes.

### 2.6. Power Calculation

A post hoc power calculation was performed in the pwr R package (v1.3.0). Assuming a mean difference and pooled standard deviation between EQ and LPT STS power estimates of 1.5 and 1.15 W/kg, respectively [23,27], 50 participants result in a power = 1.0, and, therefore, a 100% probability of detecting a mean difference of this magnitude or greater between methods.

### 2.7. Statistical Analysis

Continuous and categorical descriptive statistics are reported as mean (SD) and *n* (%), respectively. The absolute agreement between STS power methods was assessed by calculating an intraclass correlation coefficient (ICC) using the icc R package (v2.4.0) and constructing a Bland–Altman plot [42] using the blandr (v0.6.0) and ggplot2 (v3.5.2) R packages. The ICC and Bland–Altman methods are recommended to evaluate the agreement between continuous variables [43]. The unadjusted and adjusted relationships between STS power assessments and PF outcomes were assessed with Pearson product–moment correlation coefficients and multiple linear regression, respectively. Based on previous research [15], age, as a continuous variable, and sex, as a dichotomous variable, were selected as covariates a priori in the multiple linear regression models. Using the gtsummary R package (v2.1.0), standardized beta coefficients and their 95% confidence intervals are reported to compare the strengths of the associations between STS power methods and PF outcomes. All statistical analyses were completed in R (v4.4.2), using an alpha level of 0.05 for statistical significance.

## 3. Results

Overall, 94 participants were screened for eligibility, 50 participants completed all study procedures, and, due to data retrieval issues with the GymAware, 49 participants were included in the final analytic sample (Figure 1). A summary of participant characteristics is shown in Table 1.

Mean EQ and LPT STS power were 2.95 (0.98) and 5.69 (1.64) W/kg, respectively, with moderate agreement demonstrated between methods (ICC3 = 0.69 [0.51, 0.81]). The Bland–Altman plot is shown in Figure 2. Mean bias [95%CI] between EQ and LPT power was 2.73 W/kg [95%CI: 2.43, 3.04]. Additionally, there was a moderately strong positive relationship between average STS power performance and bias between methods (*r* = 0.66).

The unadjusted relationships between STS power methods are shown in Table 2. Overall, both EQ and LPT STS power were significantly associated with all PF outcomes (*p* < 0.01). Aside from HGS, EQ STS power showcased similar or marginally stronger associations with PF outcomes as LPT STS power.

The multivariate regression analyses examining the independent associations between STS power methods and objective PF outcomes are shown in Table 3. Controlling for age and sex, EQ STS power was associated with TUG (Std. β = −0.45 [95%CI: −0.74, −0.17]), UGS (Std. β = 0.37 [95%CI: 0.08, 0.66]), FGS (Std. β = 0.48 [95%CI: 0.23, 0.74]), and 400MWT (Std. β = −0.55 [95%CI: −0.77, −0.33]), but not HGS (Std. β = 0.19 [95%CI: −0.10, 0.48]). Controlled for the same confounders, LPT STS power was significantly associated with HGS (Std. β = 0.44 [95%CI: 0.18, 0.70]), FGS (Std. β = 0.40 [95%CI: 0.14, 0.66]) and 400MWT (Std. β = −0.51 [95%CI: −0.73, −0.29]), but neither TUG (Std. β = -0.26 [95%CI: −0.56, 0.04]) nor UGS (Std. β = 0.28 [−0.01, 0.57]).

The multivariate regression analyses examining the associations between STS power methods and self-reported PF are shown in Table 4. Both EQ and LPT STS power were statistically and similarly associated with self-reported total (Std. βs = 0.30–0.31), basic lower-body (Std. βs = 0.29–0.30), and advanced lower-body function (Std. βs = 0.30–0.32), independent of age and sex.

## 4. Discussion

The purpose of this study was to evaluate the agreement between EQ and LPT STS power in a sample of aging adults. Additionally, this study compared the strengths of the associations between STS power methods and PF outcomes. In line with our hypothesis, EQ and LPT STS power demonstrated limited agreement, suggesting that proposed cut-off values to identify low muscle power [23,26] should not be applied across different methodologies. Interestingly, EQ STS power seems to be similarly or more strongly associated with PF outcomes compared to LPT STS power, suggesting EQ STS power may act as a superior proxy for PF in aging adults. Given that the EQ STS power test requires relatively less time and financial resources, and less test administration expertise, its feasibility for use as a screening tool in the clinical setting should be explored further.

The finding that EQ and LPT STS power demonstrate limited agreement is clinically important, as multiple cut-off values to detect low muscle power have been proposed [23,26]. Additionally, while no clinically meaningful change in STS power has been identified [29], some research groups have proposed STS power thresholds to identify mobility limitations and disability [26,44]. Therefore, evaluating the agreement between STS power estimates of different methods is paramount for muscle power screening in clinical settings. Screening for and ameliorating low muscle power is likely to improve PF [4,14], reduce falls [16], and may promote longevity in aging adults [8,12]. The limited agreement between EQ and LPT STS power reported in this study is consistent with published normative data [23,25,27]. Indeed, Campitelli and colleagues recently reported normative STS power across the lifespan using a TENDO LPT [27]. In this study, average relative mean STS power, which was assessed in adults aged 50–80, ranged from 4 to 7.5 W/kg, which is notably higher than published EQ STS power estimates in large cohorts of similarly aged Spanish and Colombian adults [23,25], but similar to the results of this study. Given that an algorithm has been developed to detect and manage low STS power in the clinical setting [26], if multiple methods are to be used to assess STS power, method-specific cut-off values should be developed and validated in functionally and clinically diverse populations. Moreover, the feasibility of implementing STS power tests and their associated cut-off values into diverse clinical settings should be explored.

Reasons why STS power estimates differ quite drastically between methods are likely due to differences in test procedures. Specifically, EQ STS power estimates the velocity, and therefore power, of one concentric phase of the STS transition from five repeated transitions, assuming that the durations of the concentric and eccentric phases of these transitions are identical. It is likely, however, especially in aging adults, that the duration of each subsequent STS transition in a multiple-transition test (i.e., five-repetition STS test) increases over the test duration, due to the accumulation of fatigue. This, therefore, would result in an underestimation of STS velocity, which has been demonstrated previously [45]. On the other hand, LPTs precisely measure velocity [46], and therefore power, of a single concentric phase of the STS transition. Given that we assumed the same input of force (i.e., 90% of the participant’s body weight) for both tests, an underestimation of velocity will necessarily result in lower STS power values, which are clearly illustrated in this study.

Interestingly, however, EQ STS power was similarly, and in many cases, more strongly associated with PF outcomes compared to LPT STS power. It is perhaps unsurprising that EQ STS power was a stronger predictor of PF outcomes, as longer-duration STS tests are consistently more strongly associated with 6 min walk performance in both young, apparently healthy adults [47], and in adults with chronic obstructive pulmonary disease [48]. Moreover, performance in the 30 and 60 s variations of the STS test better predict performance in HGS, cognitive tasks, quality of life, and fatigue in older adults, compared to shorter-duration STS tests, such as the five-repetition STS test [49]. While this is the first study to directly compare the relationships between EQ and LPT STS power and PF outcomes in aging adults, the current study’s findings generally comport with the existing literature. Specifically, Baltasar-Fernandez and colleagues assessed the relationship between EQ STS power and PF outcomes [15], while Balachandran et al. have investigated the relationships between LPT STS power and PF outcomes using both the TENDO and GymAware LPTs [21,29]. In these studies, the unadjusted relationships between STS power and TUG performance, for example, were stronger when using EQ STS power (*r* = 0.59), compared to LPT STS power (*r* = 0.41–0.53), as the primary independent variable. However, these previous studies are limited by not measuring EQ and LPT STS power concurrently. In comparing these previous findings with the results of this study, therefore, EQ STS power and LPT STS power demonstrated unadjusted relationships with TUG performances of *r* = 0.52 and 0.38, respectively. Importantly, these differences in the relationships between STS power methods and PF outcomes persisted when adjusted for confounding variables (i.e., age and sex) and across most objective PF outcomes. Based on the results of this study, it appears that EQ STS power provides similar if not greater clarity into aging adults’ PF compared to LPT STS power, and its preferential use in the clinical setting may be justified as a result.

These findings suggesting EQ STS power, ascertained from the five-repetition STS test, is a better indicator of aging adults’ PF compared to LPT STS power may be due to a relatively large “reserve” of maximal lower-body muscle strength [50], and consequently, muscle power, when performing PF-related tasks in this population. It could, therefore, be the case that the ability to sustain muscular force and power over relatively longer periods of time (i.e., muscular endurance) is more important for from a functional perspective compared to producing maximal force and power, which is partially confirmed by the results of this current study. Indeed, while LPT STS power was more strongly associated with maximal strength (i.e., HGS) compared to EQ STS power, it was more weakly associated with ambulatory PF outcomes, which are stronger predictors of mobility limitations, falls, and mortality [51]. Additionally, PF and frailty indicators are improved to a similar degree in adults 65+ years of age randomized to resistance training (RT) or aerobic training, despite preferential increases in maximal strength after RT [52]. This suggests that maximal force production is generally not the limiting factor in maintaining PF into late adulthood and perhaps explains why LPT STS power demonstrated weaker relationships with PF outcomes compared to EQ STS power. Given that EQ STS power is validated to quantify power during both the 5 [1] and 30 s iterations of the STS test [53], and normative data have been published for both [23,25,44], future research should examine whether EQ STS power estimates agree across different test variations, and also determine if they similarly predict PF outcomes in aging adults. The results of this future investigation could better inform the selection of the most functionally relevant STS test to screen for muscle power in the clinical setting.

One limitation of this study is its relatively young and high-functioning sample of aging adults. Therefore, it is unclear whether our findings would be replicated in a sample of older and/or physically lower-functioning adults. Specific to our agreement analyses, we identified a moderately strong linear relationship (*r* = 0.66) between STS power performance and the bias between STS power methods, whereby higher average STS power was associated with greater bias. Therefore, it could be the case that these two methods agree when examining the STS power of adults with relatively low muscle power. However, given the large bias demonstrated in the current study, in addition to observable differences in normative STS power values between methods across the aging spectrum [25,26,27], it is unlikely that high levels of agreement would be demonstrated, even at extremely low levels of muscle power. Future research may aim to investigate this further, however. With respect to comparing the associations between STS power methods and PF outcomes, it is reasonable that, in an older and/or physically lower-functioning sample, with relatively less strength and power reserve, maximal power could be relatively more important for PF, perhaps resulting in similar or stronger associations between LPT STS power and PF outcomes, compared to EQ STS power. While this could be the case, longer-duration STS tests are consistently more strongly related to PF outcomes in aging adults compared to shorter-duration STS tests [48,49]. Relatedly, it is unclear whether the observed differences in the relationships between STS power estimates and PF are clinically meaningful. For example, in this study, a 1-SD greater EQ and LPT STS power was associated with a 0.37 and 0.28-SD greater UGS, which corresponds to 0.07 and 0.06 m/s, respectively. Given that the clinically meaningful difference in UGS has been cited as 0.05–0.12 m/s [54], these differences may be negligible. Nonetheless, the main questions of this study should be examined in a larger and more functionally diverse sample. Similarly, 90% of the participants in this study were non-Hispanic White, which limits its external validity. Studies with greater demographic variability should be conducted to test the generalizability of our findings. Larger studies should also be conducted to examine the moderating roles of age and sex on the relationships between STS power estimates and PF outcomes, and to determine whether method-specific cut-off values differ between patient populations. Finally, while EQ STS power shows stronger cross-sectional associations with PF outcomes compared to LPT STS power, it is unclear whether EQ STS power would be a superior predictor of “harder” clinical endpoints (e.g., mobility disability, falls, mortality). Future research should investigate the prospective associations between different STS power variations and these important clinical outcomes.

One notable strength of this study is its direct comparison between two common methods of assessing STS power in aging adults. As such, the findings from this study are pragmatic, and may inform the future implementation of screening for muscle power in the clinical setting. Another strength of this study is its use of not only objective PF outcomes but also self-reported PF outcomes, which are increasingly recognized as valuable patient data, as they provide unique information regarding a patient’s condition and abilities not otherwise captured by more objective PF outcomes [55].

## 5. Conclusions

This study compared the agreement between two STS power methods, in addition to evaluating the strengths of the associations between these methods and PF outcomes. Overall, STS power, estimated from the five-repetition STS test and from an LPT, demonstrate limited agreement. Therefore, the estimates from different STS power methods should not be used interchangeably, and the use of current and future cut-off values for low STS power should be specific to the method used. STS power, ascertained from the five-repetition STS test, is, in general, more strongly related with objective PF outcomes compared to LPT STS power. However, future research should elucidate the clinical implications of this and investigate whether EQ STS power better predicts “harder” clinical endpoints (e.g., mobility disability, falls, mortality) compared to LPT STS power. After replicating these findings in a larger, more demographically and clinically heterogeneous sample, future work should assess the feasibility of implementing STS power assessments as screening tools in the clinical setting. Identifying and referring patients with low muscle power to effective therapeutic interventions to improve muscle power (e.g., exercise interventions) [14] may prevent and ameliorate the negative individual and population health consequences of low muscle power.

## Figures and Tables

**Figure 1 healthcare-13-00905-f001:**
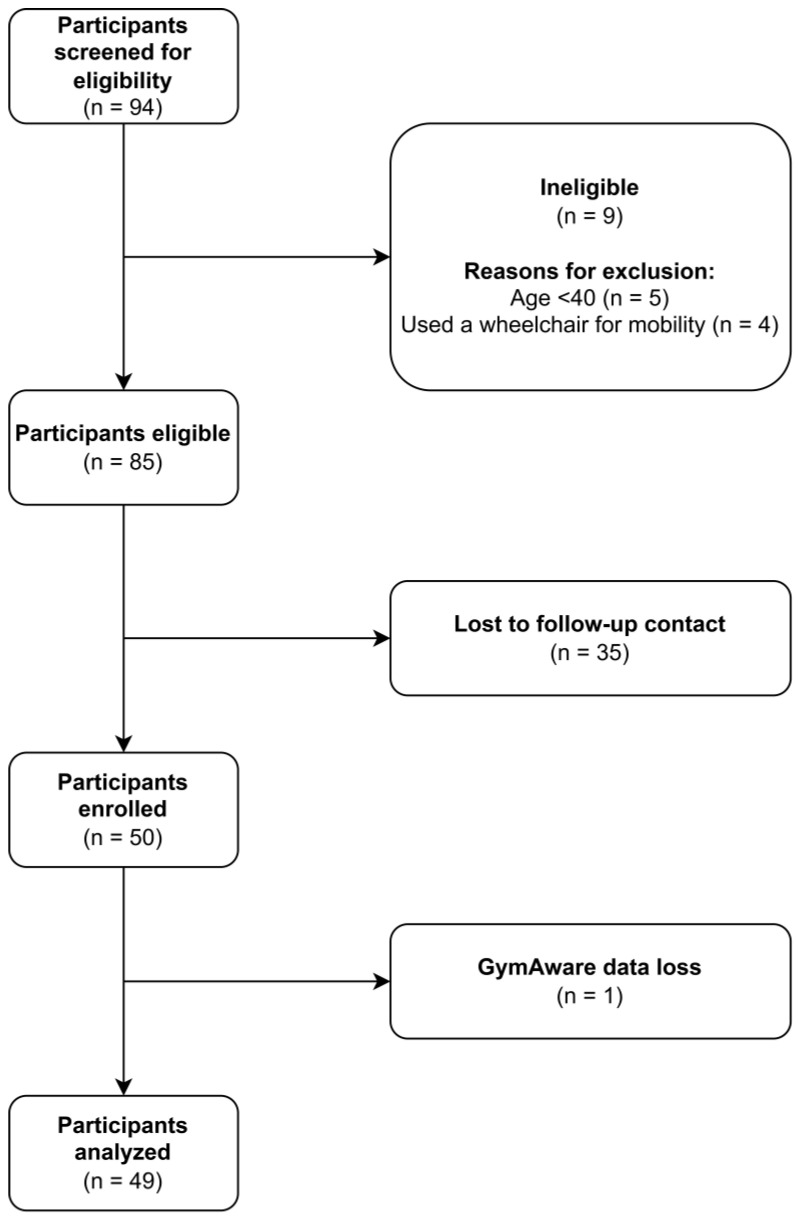
Participant inclusion/exclusion flowchart.

**Figure 2 healthcare-13-00905-f002:**
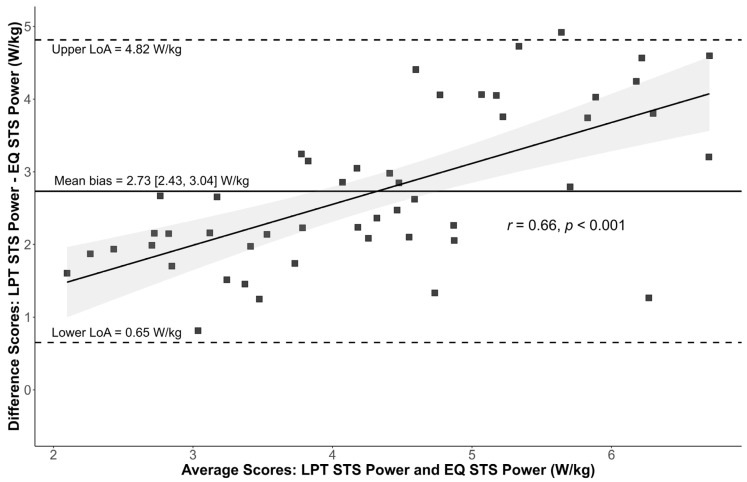
Bland–Altman plot, illustrating the mean bias and limits of agreement (LoA) between EQ and LPT STS power.

**Table 1 healthcare-13-00905-t001:** Participant characteristics.

Characteristic	N = 49 ^1^
Age, years	61 (11)
Sex	
Male	16 (33%)
Female	33 (67%)
Race and ethnicity	
Hispanic	1 (2.0%)
NH Asian	1 (2.0%)
NH Black	1 (2.0%)
NH White	44 (90%)
NH More than one race	2 (4.1%)
BMI, kg/m^2^	26.4 (5.2)
EQ STS power, W/kg	2.95 (0.98)
LPT STS power, W/kg	5.69 (1.64)
Handgrip strength, kg/kg	0.47 (0.13)
Timed up-and-go, s	8.98 (1.81)
Usual gait speed, m/s	1.37 (0.20)
Fast gait speed, m/s	2.01 (0.34)
400 m walk, s	248 (44)
LLFDI total physical function	81 (12)
LLFDI basic lower-body physical function	92 (11)
LLFDI advanced lower-body physical function	79 (15)

^1^ Mean (SD) and n (%) for continuous and categorical variables, respectively; NH = non-Hispanic, BMI = body mass index, EQ = equation, STS = sit-to-stand, LPT = linear position transducer, LLFDI = Late-Life Function and Disability Instrument.

**Table 2 healthcare-13-00905-t002:** Unadjusted relationships (*r*) between EQ and LPT STS power and physical function outcomes.

Variable	Handgrip Strength	Timed Up-and-Go	Usual Gait Speed	Fast Gait Speed	400 m Walk Test	LLFDI Total Function	LLFDI Basic Lower-Body Function	LLFDI Advanced Lower-Body Function
EQ STS power	**0.37**	**−0.52**	**0.49**	**0.61**	**−0.70**	**0.54**	**0.51**	**0.51**
LPT STS power	**0.53**	**−0.38**	**0.42**	**0.55**	**−0.67**	**0.52**	**0.48**	**0.51**

EQ = equation, LPT = linear position transducer, STS = sit-to-stand, LLFDI = Late-Life Function and Disability Instrument. Bolded correlation coefficients are statistically significant at α = 0.05.

**Table 3 healthcare-13-00905-t003:** Adjusted relationships between EQ and LPT STS power and objective physical function outcomes.

	Handgrip Strength		Timed Up-and-Go		Usual Gait Speed		Fast Gait Speed		400 m Walk Test	
	Std. β	95% CI	Std. β	95% CI	Std. β	95% CI	Std. β	95% CI	Std. β	95% CI
EQ STS power	0.19	−0.10, 0.48	**−0.45**	**−0.74, −0.17**	**0.37**	**0.08, 0.66**	**0.48**	**0.23, 0.74**	**−0.55**	**−0.77, −0.33**
LPT STS power	**0.44**	**0.18, 0.70**	−0.26	−0.56, 0.04	0.28	−0.01, 0.57	**0.40**	**0.14, 0.66**	**−0.51**	**−0.73, −0.29**

Std. = standardized, CI = confidence interval, EQ = equation, STS = sit-to-stand, LPT = linear position transducer. Bolded Std. β and 95% CI are statistically significant at α = 0.05. All models are adjusted for age (continuous) + sex (dichotomous).

**Table 4 healthcare-13-00905-t004:** Adjusted relationships between EQ and LPT STS power and self-reported physical function outcomes.

	LLFDI Total Function		LLFDI Basic Lower-Body Function		LLFDI Advanced Lower-Body Function	
	Std. β	95% CI	Std. β	95% CI	Std. β	95% CI
EQ STS power	**0.30**	**0.05, 0.55**	**0.30**	**0.04, 0.57**	**0.30**	**0.03, 0.57**
LPT STS power	**0.31**	**0.07, 0.55**	**0.29**	**0.03, 0.55**	**0.32**	**0.06, 0.58**

Std. = standardized, CI = confidence interval, EQ = equation, STS = sit-to-stand, LPT = linear position transducer, LLFDI = Late-Life Function and Disability Instrument. Bolded Std. β and 95% CI are statistically significant at α = 0.05. All models are adjusted for age (continuous) + sex (dichotomous).

## Data Availability

The raw data supporting the conclusions of this article will be made available by the authors on request.

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
