# Peer review of "Evaluating the Agreement and Associations with Physical Function Between Equation- and Linear Position Transducer-Estimated Sit-to-Stand Muscle Power in Aging Adults"

_healthcare, 2025, doi:10.3390/healthcare13080905_

Round 1
Reviewer 1 Report
Comments and Suggestions for Authors
I suggest that the authors review and make the changes I propose in the following sections:
Abstract & Introduction:
- The abstract effectively presents the study’s objectives and findings; however, I suggest the authors more clearly articulate the clinical implications of the results. It would be helpful to specify how these findings can guide the selection of STS power assessment methods in clinical practice.
- The introduction is well-structured, but it could benefit from a more detailed justification for comparing the two assessment methods. While previous studies are cited, strengthening the argument on why equation-based and LPT-based methods might yield different power estimates would add clarity.
- The study’s hypothesis should be explicitly stated, clearly outlining the expected differences between the two methods and their relative predictive value for physical function in older adults.
Methods:
- The inclusion and exclusion criteria are well defined; however, I suggest the authors provide additional details regarding the recruitment strategy and the representativeness of the sample. Were any measures taken to ensure diversity in the sample population? Providing this information would enhance the study’s generalizability.
- The description of the experimental protocol is clear, but the rationale for selecting the specific equation for STS power estimation should be explicitly stated. Clarifying why this particular equation was chosen over others available in the literature would strengthen the methodological justification.
- The statistical approach appears robust; however, I recommend that the authors provide additional justification for using the intraclass correlation coefficient (ICC) as the primary measure of agreement. Furthermore, the use of standardized beta coefficients instead of raw effect sizes in the regression analysis should be clarified. How do these choices impact the interpretation of the findings?
-The Bland-Altman plot is a valuable addition, but I encourage the authors to discuss in greater depth the implications of the observed bias. What does the systematic difference between the two methods indicate about their relative validity? Addressing this point would provide a more comprehensive interpretation of the findings.
-I suggest that the authors, if possible, include photos of the assessment tests performed. This would help the reader better understand these tests.
Results:
- The results are presented clearly; however, I suggest the authors provide a more detailed discussion of effect sizes and confidence intervals. How practically meaningful are the observed differences between methods? Including this information would enhance the interpretation of the findings beyond statistical significance.
- The correlation results should be better contextualized within the existing literature. Do these findings align with previous research? If not, what potential factors could explain the discrepancies? Addressing this would help situate the study’s contributions within the broader scientific context.
- I encourage the authors to consider conducting a sub-analysis based on sex or age groups, as muscle power assessments may vary significantly across these demographic factors. Providing these additional insights could strengthen the applicability of the findings to diverse populations.
- The tables are well-structured, but I recommend incorporating effect sizes into the regression models to improve their informativeness. This would allow readers to better evaluate the practical relevance of the observed relationships.
Discussion:
- The discussion is well-developed; however, I suggest the authors expand on the clinical implications of their findings. Specifically, it would be valuable to provide clearer guidance on how practitioners should decide between using equation-based and LPT-based STS power assessments in different clinical settings. A more explicit discussion of practical implementation could enhance the manuscript's applicability to healthcare professionals.
- The claim that equation-based STS power is a better predictor of physical function (PF) is compelling. However, alternative explanations should be explored. I recommend that the authors consider whether the nature of the movement pattern in the STS test itself might contribute to these findings. Could the repeated nature of the equation-based test better reflect endurance-related components of muscle power, which may be more predictive of functional mobility? A discussion on this aspect would provide additional depth to the interpretation of results.
- The limitations section should explicitly acknowledge the relatively small and homogeneous sample size. Given that most participants were highly functional older adults, the generalizability of the findings to frail individuals or those with mobility impairments may be limited. I encourage the authors to discuss how a larger and more diverse cohort could strengthen the external validity of their conclusions.
- Regarding future research directions, I suggest the authors emphasize the need for validation studies across different clinical settings. While the manuscript briefly mentions this, further elaboration on how feasibility studies could assess the practicality of implementing the equation-based method in real-world clinical workflows would strengthen this section. Additionally, investigating whether method-specific cut-off values should be established for different populations would be a crucial next step.
Conclusions:
-The conclusion effectively summarizes the study's key findings; however, I suggest the authors present the takeaways in a more actionable manner for clinicians and researchers. Rather than only restating the results, it would be beneficial to emphasize their practical significance, particularly in terms of how healthcare professionals can apply these findings in clinical assessments of physical function.
- I encourage the authors to include a stronger call to action regarding the standardization of STS power measurement methodologies across clinical settings. Given the limited agreement between equation-based and LPT-based assessments, it would be valuable to discuss whether separate cut-off values should be developed for each method or if efforts should focus on aligning methodologies to improve comparability.
- Lastly, the conclusion could briefly outline the next steps in research beyond feasibility studies, particularly in assessing the predictive validity of these measures for clinical outcomes such as fall risk, mobility disability, and long-term health trajectories.
Congratulations. I look forward to seeing a revised version that addresses these points.
Best regards
Reviewer 2 Report
Comments and Suggestions for Authors
This study evaluates the agreement between the equation (EQ) and linear position transducer (LPT) methods for estimating sit-to-stand(STS) power and compares the associations of these two methods with physical function (PF). The findings have significant clinical implications for the future assessment of PF in aging adults, sarcopenia screening, and fall prevention. However, there are some concerns about certain details of the manuscript. My specific comments are as follows:
- Title: The subjects in this study are aging adults, it is better to include the participants in the title to enhance the precision of paper.
- In line 14 of the abstract, the authors wrote the subjects are adults, but later in the method, they change to be aging adults. I recommend using a consistent term for the participants throughout the entire manuscript.
- In line 17, the full term "usual gait speed" before the "UGS" to reduce any confusion for readers who may not be familiar with the terminology in this field.
- In the inclusion and exclusion criteria for the participants, there is no screening for chronic musculoskeletal diseases, among others. Could this potentially affect the results of the experiment?
- According to Figure 1, there is a significant discrepancy between the number of participants initially recruited and the final number included in the study. Could the sample size meet the statistical power requirements? Providing a sample size calculation may enhance the rigor of this section.
- In section 3, the EQ STS power test consists of five consecutive trials. Is there a rest period between each trial? In the LPT STS power test, there is a 1-minute rest between trials. Could this be a potential reason for the overestimation of the LPT STS power in the final results? If there is a rest interval between the EQ STS power trials, please include this information in the manuscript.
- In Table 1, NH White participants constitute 90% of the sample, which limits the generalizability of the conclusions. This should be addressed in the limitations section of the discussion.
- In the discussion section, a discussion on the potential reasons for discrepancies between the two measurement methods should be considered. This could include aspects such as the principles of measurement or the presence or absence of intervals, as mentioned in comment 6. This would undoubtedly enrich the content of the manuscript.
- How the research results could be better applied in clinical settings, as well as a perspective on future directions?
Reviewer 3 Report
Comments and Suggestions for Authors
The study was very well designed and executed in general terms.
I will only have some endings and suggestions.
1- Why power analysis was not performed for the sample size should be analysed.
2- The limitations of the study should be stated.
3- In the first part of the discussion section, the main findings of the study should be presented and it should be explained whether the hypothesis was confirmed or not.
4- References should be organised according to the writing guide.
Round 2
Reviewer 1 Report
Comments and Suggestions for Authors
Dear authors,
Thank you for your thorough and thoughtful responses to the reviewer comments. I have reviewed the revised manuscript carefully and can confirm that you have addressed all the suggestions appropriately and in full. The revisions have significantly improved the clarity and overall quality of the paper, particularly in terms of methodological detail and clinical relevance.
Based on these improvements, I believe the manuscript is now suitable for publication.
Best regards